# Different Species Requirements within a Heterogeneous Spring Complex Affects Patch Occupancy of Threatened Snails in Australian Desert Springs

**Renee A. Rossini** [1,*], **Roderick J. Fensham** [1,2] **and Gimme H. Walter** [1] 

1   School of Biological Sciences, The University of Queensland, St. Lucia, Brisbane, QLD 4072, Australia;
    r.fensham@uq.edu.au (R.J.F.); g.walter@uq.edu.au (G.H.W.)
2   The Queensland Herbarium, Mt Coot-tha, Brisbane, QLD 4109, Australia
*   Correspondence: renee@qtfn.org.au

**Abstract:** (1) The distribution of organisms that inhabit patchy systems is dictated by their ability to move between patches, and the suitability of environmental conditions at patches to which they disperse. Understanding whether the species involved are identical to one another in their environmental requirements and their responses to variance in their environment is essential to understanding ecological processes in these systems, and to the management of species whose patchy and limited distributions present conservation risks. (2) Artesian springs in Australia's arid interior are "islands" of hospitable wetland in uninhabitable "oceans" of dry land and are home to diverse and threatened assemblages of endemic species with severely restricted distributions. Many have strict environmental requirements, but the role of environmental heterogeneity amongst springs has rarely been considered alongside conventional patch characteristics (isolation and patch geometry). (3) We quantified environmental heterogeneity across springs, and the relationship between spring size, isolation (distances to neighbours) and environmental quality (depth, water chemistry), and patterns of occupancy and population persistence of six endemic spring snail species, all from different families, and with all restricted to a single <8000 ha system of springs in Australia. To do so, a survey was conducted for comparison against survey results of almost a decade before, and environmental variables of the springs were measured. Many of the snail species occupied few sites, and environmental variables strongly covaried, so an ordination-based approach was adopted to assess the relationship between environmental measures and the distribution of each species, and also whether springs that held a higher diversity of snails had specific characteristics. (4) Each snail species occupied a subset of springs (between 5% and 36% of the 85 sampled) and was associated with a particular set of conditions. Of the six species considered in further detail, most were restricted to the few springs that were large and deep. Species in family Tateidae were distinct in having colonised highly isolated springs (with >300 m to nearest neighbour). Springs with highest diversity were significantly larger, deeper and had more numerous neighbours within 300 m than those devoid of endemic snails, or those with low diversity. (5) Although spring size and isolation affect patterns of occupancy, the six snail species had significantly different environmental requirements from one another and these correlated with the distribution pattern of each. Approaches that ignore the role of environmental quality—and particularly depth in springs—are overlooking important processes outside of patch geometry that influence diversity. These organisms are highly susceptible to extinction, as most occupy less than 3 ha of habitat spread across few springs, and habitat degradation continues to compromise what little wetland area is needed for their persistence.

**Keywords:**　　conservation/biodiversity;　groundwater;　physical　environment;　dispersal; survey/description; invertebrates; autecology; patch occupancy

## 1. Introduction

Freshwater environments are among the most altered and under-conserved global ecosystems, despite being nodes of cultural significance and endemic diversity. Freshwater surface systems that depend on groundwater, such as springs, are particularly vulnerable because increasing demands for water are leading to significant anthropogenic alteration of the aquifers that feed them [1–6]. Despite the threats they face, springs are rarely included in global assessments of freshwater ecology or conservation [1,7]. Springs in arid regions are particularly important because they provide a reliable source of water in areas characterised by water scarcity and impermanence [8–10]. They act as 'islands' of hospitable wetland in a 'sea' of aridity [11] and thus provide critical wetland environments for suites of organisms found only in springs, most endemic to a single small locality [11–14]. These severely limited geographic distributions and a history of extensive habitat loss render most of these endemic species in springs at a high risk of extinction [6,15].

In the arid and semi-arid portions of Australia, one of the world's largest actively recharging aquifers [16]—the Great Artesian Basin (GAB)—discharges naturally into an extensive system of permanent springs. These springs house a diverse assemblage of endemic organisms, most of which are aquatic invertebrates whose distributions are limited to areas <50 km$^2$ [13,17]. The habitable space within that broader area is further restricted, as the amount of spring-fed wetland in each locality is small, and few spring species occupy all springs within their distribution [18,19] or all parts of each occupied spring [20]. These restricted distributions may be caused by dispersal limitation, environmental limitation, or both. Dispersal across tracts of dry land is contingent on chance connections but is not uncommon in patchy freshwater systems in general [21], or in this system in particular [22]. In GAB springs endemic species have a limited set of environmental conditions they can tolerate, meaning not all springs within the dispersible distance are likely to provide conditions that match the environmental requirements of the species concerned. These two processes are not mutually exclusive—a species may have poor dispersal capabilities and poor intrinsic potential to colonise the sites that are reached because its environmental requirements are highly specific and not commonly met in the landscape [23]. In order to understand why these species have such narrow distributions, and what environmental changes may imperil their existence, we need to understand patterns of patch occupancy and the environmental correlates of occupancy for each species.

In springs, as in other freshwater systems, dispersal limitation appears to play a primary role in limiting the distributions of species, as well as influencing patterns of spring occupancy within each species' range [24,25]. Most species endemic to Australia's GAB-fed springs are sensitive to desiccation despite their occurrence in an arid context [26,27], and therefore have poor abilities to disperse actively across the dry country that persists between each spring wetland [28]. Dispersal appears to be facilitated by different processes at different spatial scales. For example, the spring snail *Fonscochlea accepta* disperses over short distances (<300 m) by actively traversing the wet ground that persists between spring vents [22], some appear to be transported across greater distances (<3 km) attached to larger animals (phoresy) [19], and populations in the same drainage basin are mixed indiscriminately during infrequent but intense flooding events [29]. The ability of each species to disperse across the landscape, and the channels most commonly used, differ across taxa; for example endemic amphipods (e.g., *Wangiannachiltonia guzikae*) appear to be poor dispersers, with molecular evidence suggesting populations within close proximity are highly divergent, whereas endemic ostracods and snails (e.g., *Ngarawa dirga* and *F. accepta*) appear to disperse readily across springs within the same drainage basin because spatially isolated populations show little divergence across springs separated by large distances and little flow connectivity [28].

The stabilising influence of groundwater in springs often leads to the assumption that each spring-fed wetland within a particular area, or which is fed by the same source, provides the same environmental conditions [30]. Such assumptions of homogeneity are particularly common in metapopulation and metacommunity approaches to studying patchy or insular systems like desert springs [31] but are not ubiquitous across all studies [32]. Whilst such assumptions may be valid in systems where a previously contiguous and homogenous landscape has been fragmented into patches (e.g., fragmented forest landscapes [33–39]), the "quality" of a patch can be as important in dictating patterns of occupancy as patch geometry and isolation in systems where the conditions vary across patches [40–45]. In such cases, the match between an organism's species-specific environmental requirements and the environmental conditions in a locality are likely to be as important as dispersal-oriented processes (as anticipated by autecological theory [46] rather than community theory) and their interaction with the distribution or size of suitable localities. Environmental conditions other than size or connectivity have been shown to be the key determinant of diversity in a range of springs systems, from limestone systems to desert spring complexes [41,42,47–51]. Further, the relative importance of the environmental requirements of organisms in dictating patterns of distribution and population connectivity appears, as with dispersal mechanisms, to be highly species specific [25,52].

Springs fed by the GAB are unlikely to be environmentally homogenous because of the unique hydrogeology and topography of each spring. The interaction between groundwater and local geological context creates different water chemistry environments across springs in a complex [48]. For example, the particular stratigraphy underlying each spring, or the accumulated salts in the sediments from which a spring emerges, strongly affects the ionic composition of the water that fills them [48,53]. Microhabitat diversity within springs, particularly depth and flow regimes, strongly influence spring assemblages [20,48,54], with elements appearing related to a spring's flow rate [55] and age [56]. More specifically, deep spring pools with strong connections to groundwater maintain more stable conditions, whereas shallow areas and springs that lack pools are more variable environmentally [20,48,57]. Such environmental variance directly affects the distribution of freshwater organisms, including those in springs. For example, few species of snail endemic to springs are able to persist for more than 6h when left dry [27,58] (a situation that occurs daily in shallow spring areas in GAB springs [20]), and species whose distributions are restricted to deep pool areas suffer increased mortality when forced to persist under temperature regimes, pH or conductivity conditions that typify shallow areas [27]. Therefore, the size or depth of the pool within each spring, and the conductivity and pH of that pool, are potentially just as influential ecologically as the isolation of a spring.

We assessed patterns of environmental heterogeneity within a GAB springs complex with high endemic diversity, and how these related to patterns of occupancy and persistence in six species of spring snail endemic to the area. We had three aims: (1) to describe patterns of occupancy of six species from three different families (*Glyptophysa* sp., *Gyraulus edgbastonensis*, *Jardinella acuminata*, *J. jesswiseae*, *J. edgbastonensis* and *Gabbia fontana*) and how they have changed since previous surveys in 2006–2008, (2) to assess the extent of environmental heterogeneity across springs, and (3) to assess the way in which each snail species responds to this environmental heterogeneity.

## 2. Methods

### 2.1. Site Description

The Pelican Creek springs complex is located in central Queensland, Australia (Figure 1). The site was chosen as it has the highest concentration of endemic species of the GAB spring system, and is one of few locations where endemic species from different families persist in sympatry. The northern portion of the complex is enclosed within the Edgbaston conservation reserve, managed by Bush Heritage Australia. The springs of the Pelican Creek complex are spread across a north to south axis, with the northern springs at the base of a rocky escarpment (spring code begins with 'N', latitude −22.725 to −22.721), the central springs mostly within a large clay pan and scald (spring code begins

with 'E', latitude −22.725 to −22.74) and the southern springs within, or in proximity of, the large ephemeral Lake Mueller (which drains into the nearby Aramac creek) (spring code begins with 'S', latitude −22.74 to −22.76). The complex continues to the south of the Lake, into an adjoining property outside of the conservation reserve that contains additional endemic species.

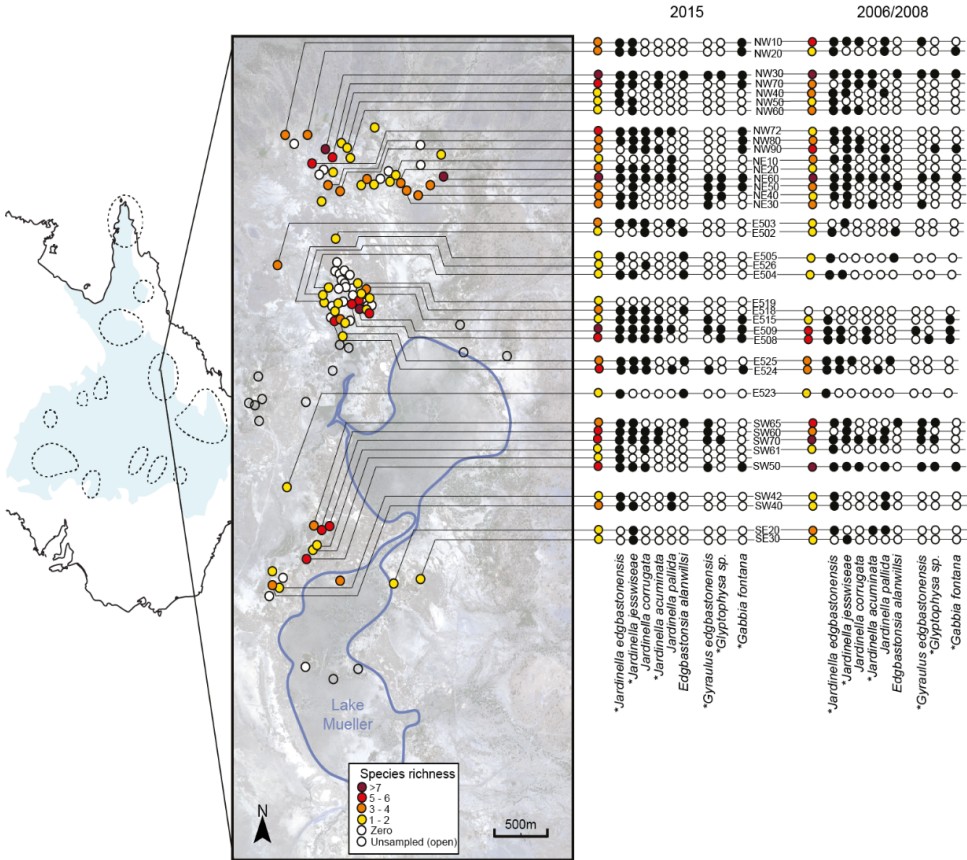

**Figure 1.** All springs in the Edgbaston Reserve portion of the Pelican Creek springs complex, showing springs as circles on a satellite image of the landscape. Springs are colour coded based on the number of species found in each spring in the 2015 sampling period. On the right, for both 2015 and the earlier 2006/2008 sampling period (Ponder et al. [18]), which species occupied each spring at that time (black = occupied, white = absent) for each species of snail endemic to the complex. Springs that were never found to contain snails are indicated, in white, on the map. The map of Australia has the major spring complexes of the Great Artesian Basin delimited with dotted lines.

These springs all have shallow open water pools of a limnocrenic morphology [59]. The complex, as a whole, comprises ~145 springs, with 113 of those within the reserve. Eighty-five of the springs within the reserve were sampled during this study. All 37 springs sampled in extensive surveys conducted by the Australian museum for conservation purposes in 2006 and 2008 [18] were revisited, as well as an additional 48 springs spread throughout the complex that were accessible in 2015 (but were not so in 2006 and 2008). We endeavoured to sample the same springs as the previous surveys of Ponder et al. [18] by matching the site names to those included in the current springs database (held by the Queensland Herbarium as part of the Lake Eyre Basin springs assessment (LEBSA) [60], on the assumption that these names are consistent.

*2.2. Study Species*

Nine species of snail are endemic to the Pelican Creek complex: *Gyraulus* (*Gy*.) *edgbastonensis*, an undescribed species of *Glyptophysa* sp. considered to be endemic to the Pelican Creek springs

complex [61], *Gabbia* (*Ga.*) *fontana* [62], *Jardinella acuminata*, *J. corrugata*, *J. edgbastonensis*, *J. jesswiseae*, *J. pallida* and *Edgbastonia alanwillsi* [63,64]. All nine species were collected during the surveys of all endemic fauna conducted by Ponder et al. [18] and were included in the re-survey conducted here. Six of the nine species were included in further detailed analyses of environmental limits. These were chosen to ensure that representatives from each family were included (*Glyptophysa* sp., *Gy. edgbastonensis* (Planorbidae) and *Gabbia fontana* (Bithyniidae)), and where there were multiple species within a family (Tateidae), those species with different patterns of distribution from one another were included (*J. edgbastonensis*, *J. acuminata* and *J. jesswiseae* have all been demonstrated to have different environmental associations [20] and tolerances [27] from one another).

### 2.3. Environmental Variables

Four broad types of environmental covariates were selected: spring size, pool characteristics, water chemistry and connectivity (for more detailed information on environmental variance in these springs see [20]). Spring size was measured using two methods. Quickbird images of the Pelican Creek complex from 2013 were manually mapped to estimate wetland area. All spring vent locations were marked onto the image using GIS, and ImageJ software was used to trace the edge of the darkened vegetated area around each vent. This measurement is referred to as 'Vegetated area'. On-site estimates of the amount of the wetted area of a spring were also made. When each spring was visited for sampling, the length of the spring at its longest axis and the width at its widest axis were measured. The spring was considered to extend into any area where there was moist sediment. These measures were used to construct an estimate of 'Wetted area' by calculating the area as a rhombus. 'Pool area' was estimated using the same methods as the on-site estimates of wetted area, but the greatest length and width of the area where water surrounding the spring vent was >5 mm deep (i.e., there was some standing water, not only wetted sediment) was measured. Spring pool maximum depth was measured by taking 10 replicate depth measurements across the spring, including at least one from the area that appeared to be the spring vent (i.e., signs of water bubbling, vent associated vegetation). Only the maximum was used for analysis.

The water chemistry of springs was taken from the long-term data set of the pH and conductivity of the water at the vent of each spring at Edgbaston held by the Queensland Herbarium [60]. As the pH and conductivity of the spring vent can fluctuate through seasons and years, this analysis used the average pH and conductivity of water measured at the spring vent on at least three separate occasions prior to and including 2014. These measures were taken across multiple days within the 2014 survey periods and at other times since 1998. Measures of pH and conductivity were not available for 56 springs out of the 85, so only a subset of the full dataset could therefore be analysed in regard to these variables.

Measures of spring connectivity were made using a number of methods. Accurate information regarding channels, water paths and elevation on Edgbaston is not available, so accurate measures of potential connections between springs based on water movement were not possible. Therefore, measures are based on the Euclidean distance between springs. A matrix of the Euclidean distance between all springs was constructed using the GPS locations of each spring vent. This was used to calculate the average distance to all springs and to the nearest five neighbours. Two additional measures of connectivity were included. For each spring, the number of other spring vents connected to the focal spring via continuous wetland at the time of sampling in 2015 was counted. Spring wetland outflows and associated wetland areas can be highly transient [20,55] and past connections may be overlooked in this single year of sampling, so the number of spring vents within 300 m was included as an additional variable.

### 2.4. Snail Survey Method

All springs were located onsite using a handheld Garmin GPS pre-loaded with the locations for all springs on the Edgbaston property ascertained from the LEBSA database [60]. All sampling was conducted in a four-day period in April 2015. Sampling for endemic snails followed recommended

methods [65]. Five replicate 10 by 10 cm scoops of vegetation and sediment were taken in five different locations in the shallow outflow of each spring, and five in the deep pool area of each spring. If a spring was small (<9 m$^2$) and/or had no deep pool area, only five samples were taken. Scoops were taken using a modified aquarium net fitted with a 1 mm mesh (these methods differ slightly to those of Ponder et al. [18] in that the previous survey used a kitchen sieve, rather than an aquarium net). Each sample was emptied onto a sorting tray with a few millimetres of water from the spring of origin, and left for 2 min for sediment to settle and animals to begin moving. Each tray was searched for 2 min and as many snails as possible were collected into a separate dish. These samples were checked under a dissection microscope and identified to species level. A sample of ten individuals of each species in each spring was collected and stored in 100% ethanol, and kept as a voucher collection.

## 3. Data Analysis

Most species occupy few springs (<10) and many springs contained no snails. Additionally, environmental variables were strongly correlated and extensive time series were unavailable. These points limited options for statistical analysis [66]. Conventional binomial modelling was trialled but the low number of sites occupied rendered this style of analysis of little use. Occupancy models were not possible due to the limited number of time points. Therefore, a descriptive approach that used visualisation on bi-plots and ordination was chosen, and all hypotheses focused on the dissimilarities across springs, and across the characteristics of springs occupied by each species.

Most of the analyses conducted here are identical to those recommended by Borcard, Gillet, and Legendre [67] and Legendre and Legendre [68] for use within the 'vegan' package for community ecology [69]. However the conventional logic recommended within the 'vegan' package was inverted—instead of creating ordinations based on the 'communities' of species and their relative abundances, and assessing their relationship with environmental predictors, the ordination was based on the dissimilarity in environmental variables across springs (i.e., points are sites whose dissimilarity is calculated using their environmental characteristics) and the occupancy of each species was projected onto this. The key drivers of dissimilarity across springs were analysed using the 'envdist' function within the 'vegan' package. The overlap across species (regarding the environmental conditions within springs they occupied and persisted within) was assessed using the centroids of occupied springs in ordination space. All centroids were calculated using the 'ordispider' function and standard error ellipses calculated and fitted using the 'ordielipse' function, both within the 'vegan' package [69] executed in R Studio.

In making comparisons, therefore, we considered how patterns of occupancy in each species was related to each environmental variable using bi-plots, and also how they occupied "environmental ordination space" by looking to where the "average" (i.e., the centroid and its standard error) of occupied sites lays, how similar these are across species (i.e., if the centroids are in the same place), and the proportional amount of "ordination space" occupied (i.e., a species that occupies many springs with dissimilar environmental characteristics occupies a larger portion of "environmental ordination space").

We also endeavoured to test whether springs with different species richness had different environmental conditions. First, the environment-based springs ordination was coded with four diversity categories: no species, species richness ≤2 composed of only widespread species (low), species richness between 3–6 composed of widespread species and at least one species that occupies few springs (moderate) and species richness >6 comprising at least 3 species that occupy few springs (high). As there appeared to be differences across diversity categories, parametric tests of the differences across sites with different species richness were also conducted. Few springs satisfied the "high" conditions, so data were split into a different set of four bins: 0, 1, 2, 3–4 and >4 species present. Random springs were removed from each bin until all bins had an equal sample size of twelve. For each of the 10 environmental variables measured, an ANOVA was used to assess whether there was a difference in the mean value of each metric across the bins (i.e., are springs with >4 species significantly larger than all others). Individual pair-wise differences were assessed using a Tukey HSD post hoc test. Whether

the data for each metric met the assumptions of ANOVA was tested using the Cochrans C test ('C.test' function within the 'GAD' package) and the Shapiro-Wilks test ('shapiro.test' function within the 'nortest' package) and, if needed, data were transformed using log transformation.

## 4. Results

### 4.1. Patterns of Occupancy and Changes since 2006–2008

In the 2015 survey, only 70% of springs sampled within the Edgbaston Reserve section of the Pelican Creek complex were occupied by any snail species, and occupied springs were spread across the reserve (Figure 1). No species was found in all of the occupied springs (Figure 1), with most of them in <20% of springs (*Gy. edgbastonensis*, *Glyptophysa* sp., *J. acuminata* and *Ga. fontana*) (Figure 1). *Jardinella jesswiseae* and *J. edgbastonensis* occupied the most springs (42% and 50% respectively) (Figure 1).

In the sub-set of 37 springs included in the survey of Ponder et al. [18], all species experienced extirpations and colonisations (Table 1). *Gabbia fontana* occupied considerably more springs in 2015 (Table 1), having colonised six springs it was not found in previously: five from the north-west (NW20, NW70, NW72, NW80 and NW10) and one from the centre (E524). However, it did experience one extirpation (NW90). *Gyraulus edgbastonensis* occupied the same number of springs in 2015 but they were not the same springs. It experienced two extirpations in the far north and south (SE10 and NE10), and two colonisations in the centre of the complex (E515 and E524). Three springs occupied by *Glyptophysa* sp. remained occupied in 2015, an extirpation occurred in SW65 and two new springs were colonised (NE60 and E509). *Jardinella acuminata* occupied more springs in 2015 and colonised six springs from across the complex (NW90, SE10, SW60, NW72, E515 and E508), but suffered extirpations in SE20 and NE30. Most springs occupied by *J. jesswiseae* in 2006/2008 were also occupied in 2015, with five colonisations in the north-west (NW50, NW20), south-west (SW40) and centre of the complex (E518, E515). *Jardinella edgbastonensis* suffered four extirpations (NE10, E502, SE20, NW60) and colonised seven springs, primarily in the north (NE20, NE30, NE60, NW72) and centre of the complex (E503, E518) (with one in the south, SW60).

**Table 1.** Six species of spring snails and their status with respect to the 2006 and 2008 survey of Ponder et al. (2010) survey of 2006–2008 and the results of the current survey (2015). The table shows the number of springs occupied by each species (out of 85 springs sampled). For each snail species is given the number of springs that still have a population (labelled 'Same'). Additionally, the number of springs is given from which that species has been extirpated or colonized by it since 2008.

| Snail Species | 2006–2008 | 2015 | Same | Extirpation | Colonisation |
|---|---|---|---|---|---|
| *Gabbia fontana* | 7 | 12 | 6 | 1 | 6 |
| *Gyraulus edgbastonensis* | 10 | 10 | 8 | 2 | 2 |
| *Glyptophysa* sp. | 4 | 5 | 3 | 1 | 2 |
| *Jardinella acuminata* | 7 | 11 | 5 | 2 | 6 |
| *Jardinella jesswiseae* | 25 | 28 | 23 | 2 | 5 |
| *Jardinella edgbastonensis* | 28 | 31 | 24 | 4 | 7 |

### 4.2. Patterns of Environmental Heterogeneity

The springs of Edgbaston fall into a range of sizes (<1 m$^2$ to 10,000 m$^2$) (Figure 2), and vary substantially in their pool characteristics (no pool, to pools over 100 m$^2$ and 40 cm deep), pool conductivity (500 to 2132 uS/cm) and pH (6.75 to 9.62) (Figure 2). Springs of all sizes are found across the complex, though springs >1000 m$^2$ are not present in the central region (Figure 2). Estimates of vegetated area taken from Quickbird images and the on-ground estimate of wetted area differ because the area of wetted sediment often extends past the extent of the vegetated area, and estimates of vegetated area from Quickbird images for springs <100 m$^2$ were impossible (and are thus 0 m$^2$ when they are, in fact, vegetated) (Figure 2). The best, but by no means most accurate, predictor of the

relationship between these measures was a power relationship (wetted area = 0.65 × vegetated area 1.26, $r^2$ = 0.61).

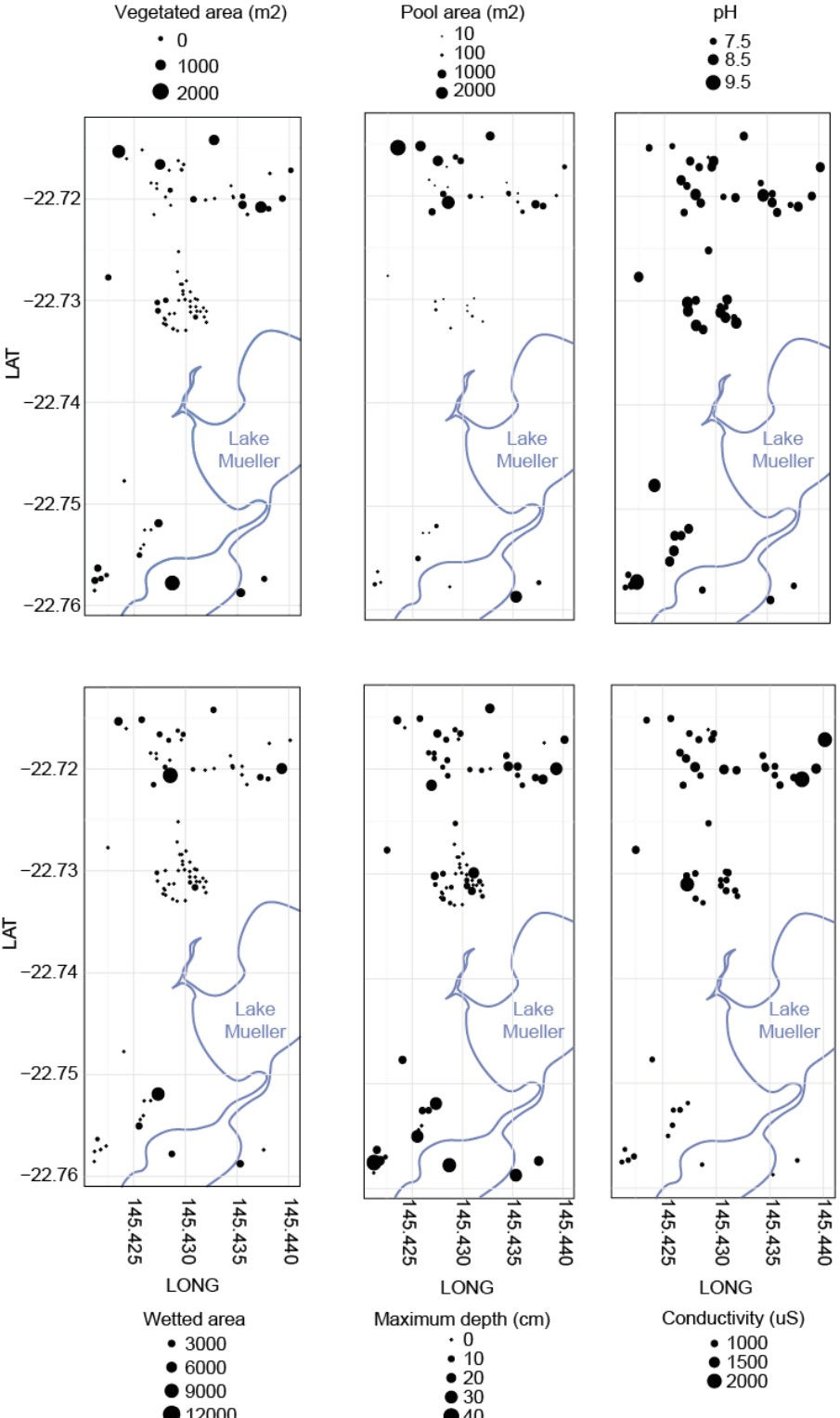

**Figure 2.** All springs sampled at Edgbaston mapped according to six environmental variables in three broad categories: size (left) as vegetated area (m$^2$, top) and wetted area (m$^2$, bottom), pool (centre) as pool area (m$^2$, top) and maximum pool depth (mm, bottom) and water chemistry (right) as pH (top) and conductivity (uS, bottom).

Close to a third of springs (27.3%) lack a pool (Figure 2), and most pools are small (mean pool size (Q1, Q3) = 334 m$^2$ (6, 460)). Springs with large pools (>2000 m$^2$) (Figure 2, centre) are primarily found in the north of the complex (Figure 3). Large pools are not necessarily deeper (R$^2$ = 0.08), nor are they only found in springs with large vegetated area (both cases R$^2$ = 0.5). Overall, spring pools at Edgbaston are shallow (all are <80 cm maximum depth), and the deeper springs (>10 cm) are found throughout the complex (Figure 3). There is a latitudinal conductivity gradient, with springs in the far south having lower conductivity than those in the north (possibly due to the influence of the large ephemeral Lake Mueller in the south-east of the property, see Figure 3), but the same cannot be said for pH (Figure 2).

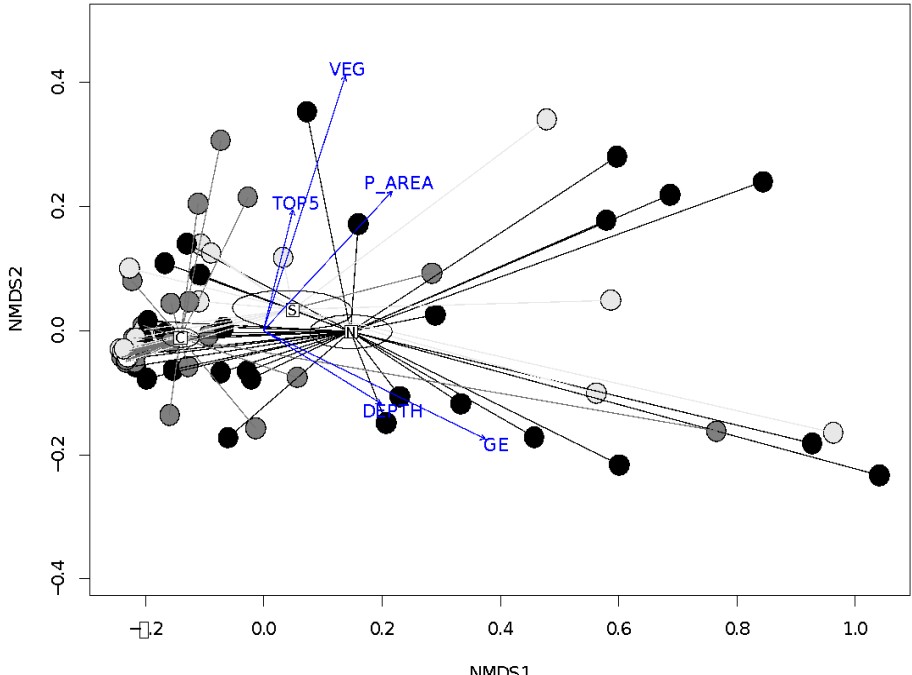

**Figure 3.** nMDS of a distance matrix calculated from the environmental characteristics of all springs at Edgbaston, colour coded for the latitudinal area of the complex they are within (black = north (latitude −22.71 to −22.725), dark grey = central (latitude −22.726 to −22.74) and light grey = south (latitude −22.75 to −22.76). Overlayed are the centroids (label position) and their standard errors (coloured ellipses). In blue, the significant environmental drivers of the spread of points in ordination space are displayed and are labelled by codes as follows: VEG = vegetated area (m$^2$); TOP5 = average distance (m) to the five closest springs; P_AREA = pool area (m$^2$); GE= wetted area (m$^2$); DEPTH = maximum depth (mm).

In ordinations of the similarities of springs based on their environmental characteristics, the springs do not form clusters with particular conditions, or clusters based on their geographic position within the complex, but are spread across a continuum. There are springs from each region spread throughout ordination space (Figure 3). The centroid of springs from the north and the south are closer and thus are similar to one another, and have overlapping standard errors, but the centroid for springs from the centre is distant from these, suggesting that springs from the central scald are, on average, dissimilar to those from the north and south. This is due to the vast majority of springs in the centre of the complex being small and shallow. However, there are springs from the centre that have large pool areas, large vegetated areas and greater depths, just as there are in the north and south. The environmental variables that significantly affect the spread of springs across ordination space are vegetated area and the wetted area (both $p = 0.001$), pool size and maximum depth (both $p < 0.05$) and average distance to the closest five springs ($p = 0.05$).

### 4.3. Relationship between Occupancy and Spring Characteristics

Each snail species occupied springs characterised by different environmental characteristics (Figures 4 and 5). Within the Planorbidae, *Gy. edgbastonensis* and *Glyptophysa* sp. were found in the largest and deepest springs, though they differ in that *Glyptophysa* sp. was found on both sampling occasions in the largest, deepest springs with the largest pools (Figure 4A,B) Despite these common requirements, they differ from one another in that *Gy. edgbastonensis* was found in springs with pools >1 m$^2$ with maximum depth >2 cm, and *Glyptophysa* sp. was found in springs with pools ~100 m$^2$ or larger, and depth of 10 cm or deeper (Figure 4A,B)). Both species were absent from any spring with average distance to the nearest five neighbours >300 m (Figure 4A,B). The only species of bithyniid, *Ga. fontana*, occupied similar springs to those inhabited by *Gy. edgbastonensis* (Figure 4C), with the key differences being that springs occupied by *Ga. fontana* are generally larger than 100 m$^2$ wetted area.

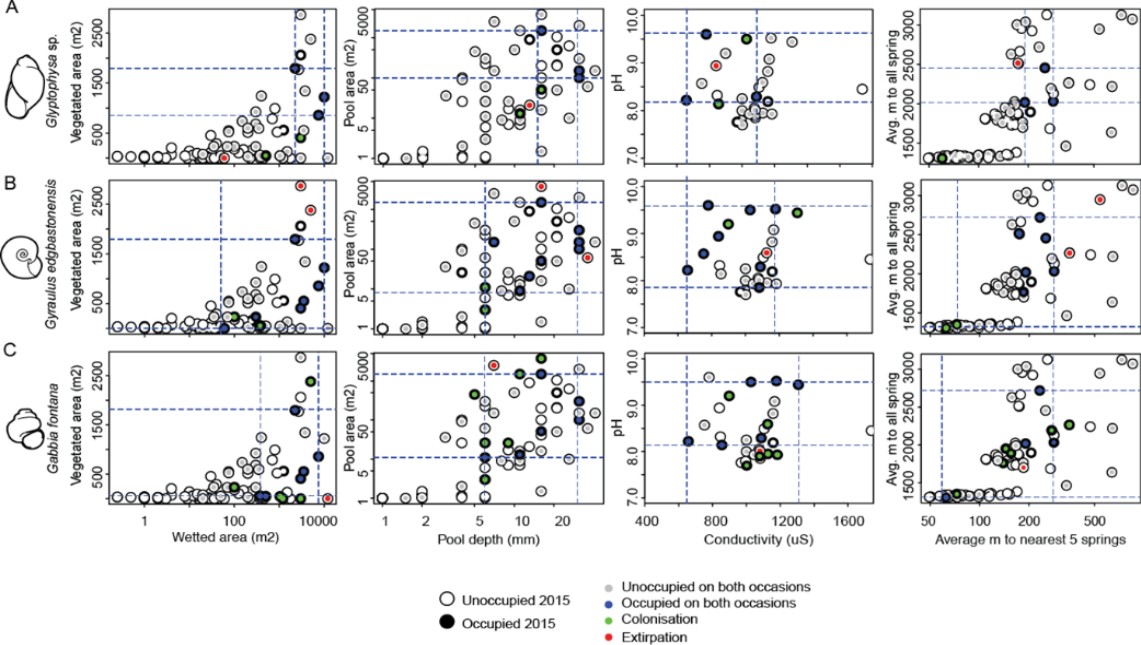

**Figure 4.** Bi-plots of the four types of environmental variables considered here: spring size (left, vegetated area and wetted area), pool characteristics (centre-left, pool area and maximum pool depth), pool water chemistry (centre-right, pH and conductivity) and isolation by Euclidean distance (right, average distance to closest 5 springs and to all springs). Overlaid on these bi-plots is the presence and absence (black and white) of three of the six focal species (**A**, *Glyptophysa* n. sp.; **B**, *Gyraulus edgbastonensis*; **C**, *Gabbia fontana*) and, for springs where samples are available, whether that spring is a site where the species has been recorded on both occasions (blue) or neither occasion (grey), or has been a site of a colonisation (green) or extirpation (red). Grey and blue lines represent the maximum and minimum value for all springs ever occupied (grey) and those occupied on both sampling occasions (blue).

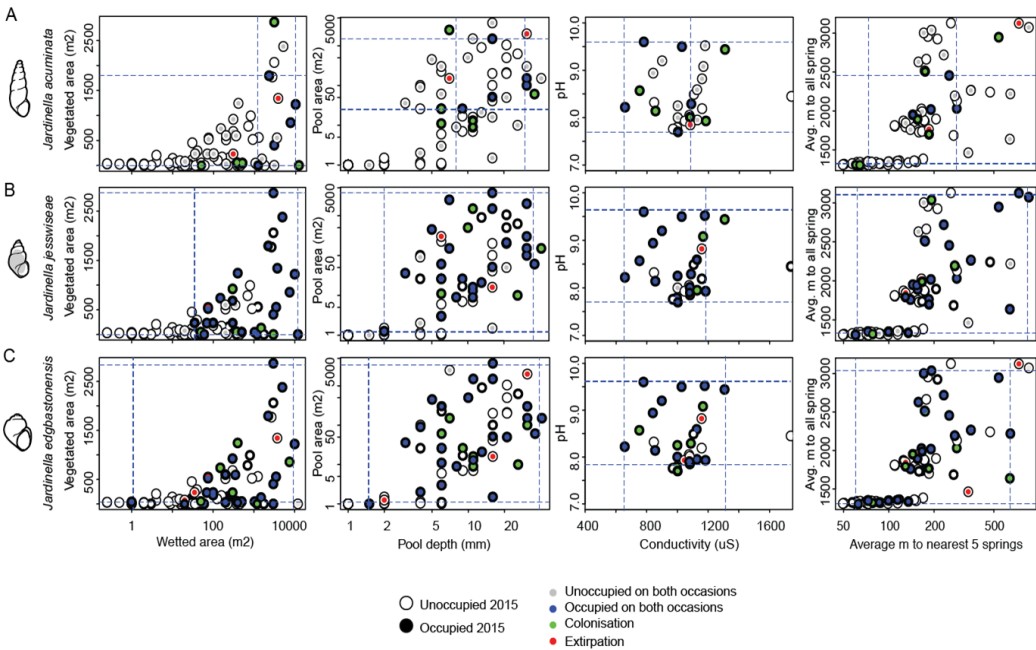

**Figure 5.** Bi-plots of the four types of environmental variables considered here: spring size (left, vegetated area and wetted area), pool characteristics (centre-left, pool area and maximum pool depth), pool water chemistry (centre-right, pH and conductivity) and isolation by Euclidean distance (right, average distance to closest 5 springs and to all springs). Overlaid on these bi-plots is the presence and absence (black and white) of three of the six focal species (**A**, *Jardinella acuminate*; **B**, *Jardinella jesswiseae*; **C**, *Jardinella* edgbastonensis) and, for springs where samples are available, whether that spring is a site where the species has been recorded on both occasions (blue) or neither occasion (grey), or has been a site of a colonisation (green) or extirpation (red). Grey and blue lines represent the maximum and minimum value for all springs ever occupied (grey) and those occupied on both sampling occasions (blue).

The three species of *Tateidae* are similar to one another, in that they all occupy springs across the gradient of isolation (i.e., they are found within springs that have few springs within 300 m, in springs that have many others nearby, and in springs with average distances to the nearest neighbour >300 m) (Figure 6). However, they differed considerably in the size and pool characteristics of springs they occupied. *Jardinella acuminata* was only found in the largest (ground estimated area >100 m$^2$) springs with the largest and deepest pools (>1 m$^2$ pool area, >5 cm maximum pool depth) (Figure 5A). *Jardinella jesswiseae* was found in smaller springs (10–100m$^2$ ground estimated area) with shallower pools (>2cm maximum depth) (Figure 5B). *Jardinella edgbastonensis* was the only species of all six focal species to occupy the smallest (<10 m$^2$) springs with little (<1 m$^2$) or no standing pool of water or with very shallow pools (<1cm maximum depth) (Figure 5C). *Jardinella jesswiseae* and *J. edgbastonensis* were the only species found on both sampling occasions in springs with the greatest distance to all other springs or to the nearest five (Figure 5B,C).

In the subset of 37 springs for which there is time replication, *Glyptophysa* sp. suffered extirpations in the smallest spring with one of the smallest shallowest pools, and the locations of colonisations were all in springs with the shortest distance to neighbours (Figure 4A). In *Gy. edgbastonensis*, the two extirpations were from two very large isolated springs and all colonisations were in springs with the lowest distance to neighbours (Figure 4B). For *Ga. Fontana*, the single extirpation was from a very large spring with a large pool of average depth (Figure 4C). *Jardinella acuminata* was found, in both surveys, in springs that are larger with larger, deeper pools and moderate distances to neighbours, though there are no obvious relationships across sites they have colonised or been extirpated from since the last survey (Figure 5A). There are no obvious patterns for J. jesswiseae and *J. edgbastonensis* (Figure 5B,C).

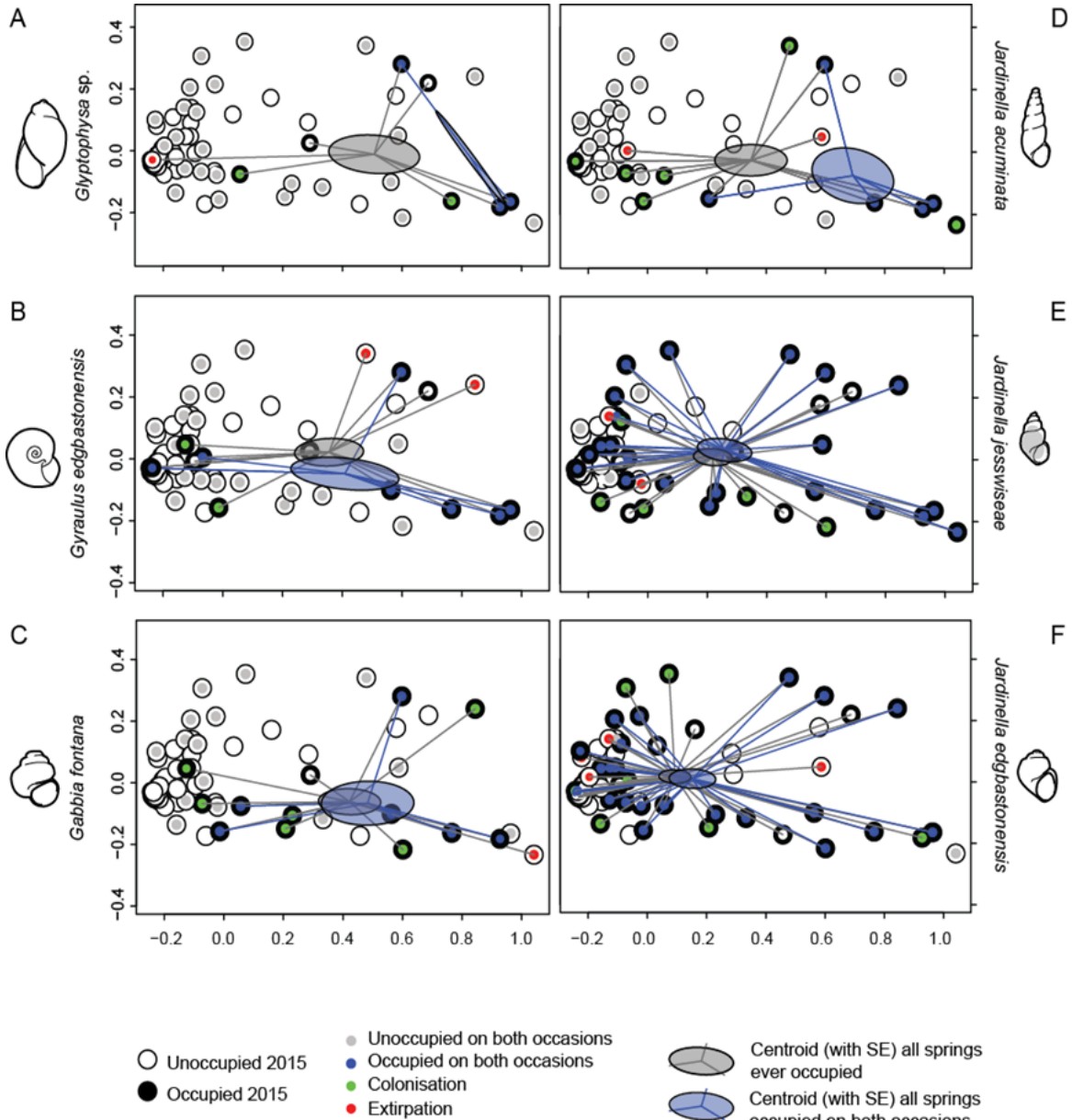

**Figure 6.** nMDS ordinations for each of the six focal species (**A–F**) showing all springs sampled at Edgbaston based on the dissimilarities in their environmental characteristics colour coded for whether the species was present or absent in the 2015 survey (black or white) and for springs where samples are available, whether that spring is a site where the species has been recorded on both occasions (blue) or neither occasion (grey), or has been a site of a colonisation (green) or extirpation (red). Ellipses represent the standard error of the centroid of either all springs ever occupied (grey) or all springs occupied on both sampling occasions (blue).

These patterns combine to mean that each snail species occupies different areas of ordination space when projected onto the environmental dissimilarity ordination plot across springs (Figure 6). Springs occupied by *Glyptophysa* sp. and *J. acuminata* are the most limited—in both species, all springs occupied on both occasions are similar to one another, and cluster together on the far right, meaning they have the largest vegetated areas and largest and deepest pools. These springs have centroids whose standard errors do not overlap with those occupied by any other species (Figure 6A–F, respectively). *Gyraulus edgbastonensis* and *Ga. fontana* occupy springs with similar characteristics (i.e., the centroids

and errors for these two species completely overlap, Figure 6B,C, respectively). *Jardinella jesswiseae* and *J. edgbastonensis* also occupy springs with similar environmental characteristics (their centroids and errors overlap with each other), however unlike J. jesswiseae, *J. edgbastonensis* occupies springs that are dissimilar to those occupied by the other species, in that the centroid and error for this species overlaps with no other species apart from *J. jesswiseae* (Figure 6E,F).

## 4.4. Relationship between Diversity and Spring Characteristics

Each diversity category (see methods) was found in springs within a particular part of the ordination space (Figure 7). Springs unoccupied by any species are completely dissimilar to those that are occupied by more than six species (Figure 7), with no springs in this "high" diversity category being found within the 90% confidence interval encapsulating springs with no snails. Unoccupied springs are significantly smaller, shallower, and have fewer springs connected by wetland (Table 2). High diversity springs are also dissimilar to those that support only *J. jesswiseae* and *J. edgbastonensis*, or springs with at least one species that occupies fewer than 15 springs (i.e., the centroids of these treatments do not overlap with the confidence interval of springs that house no endemic snails), but some springs occupied by common species are environmentally similar to unoccupied springs (they occur within the standard error centroid for unoccupied springs). High diversity springs, and particularly those that contain at least three species that occupy few springs, are significantly larger and deeper than all other springs and have a higher number of springs within 300 m connected by wetland (Table 2).

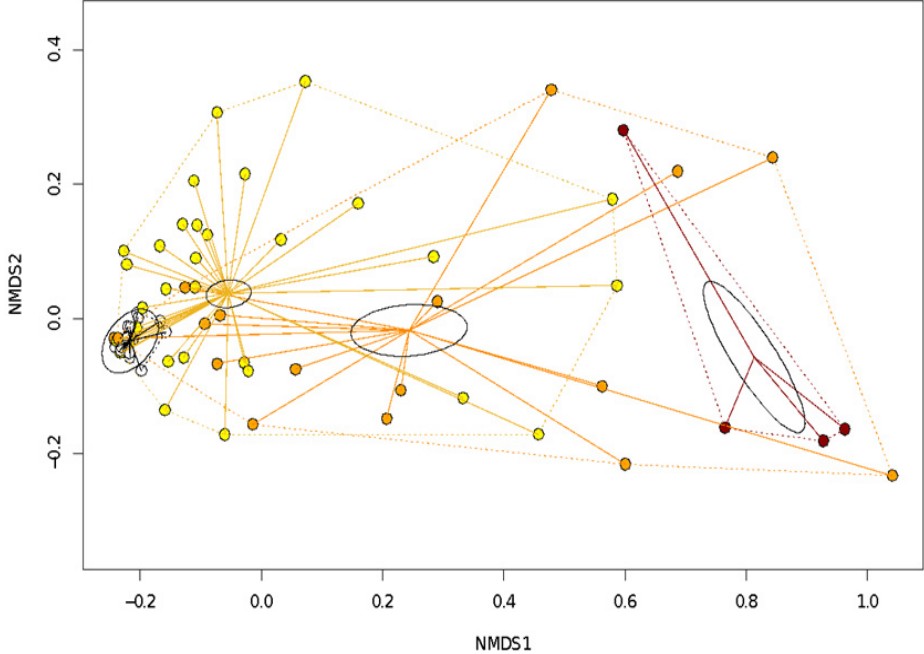

**Figure 7.** nMDS ordination showing all springs sampled at Edgbaston based on the dissimilarity in their environmental characteristics colour coded by four categories of species diversity: no species (white), species richness ≤2 composed of only widespread species (yellow), species richness between 3–6 composed of widespread species and at least one species that occupies few springs (orange) and species richness >6 comprising at least 3 species that occupy few springs (red). Ellipses for each of the diversity category represent the standard error of the centroid and for springs with no species (white) represents the 95% confidence interval.

**Table 2.** The relationship between species richness and patch characteristics for Edgbaston. Average value and standard error for springs within each species richness category are shown for each complex alongside results of individual ANOVA (n = 12 in five categories of species richness) for each factor. No significant overall effect is denoted with "ns". Where a significant overall effect was found the *p*-value is presented as are the results from Tukey HSD post hoc tests used to investigate individual pair-wise differences. Factors unable to be analysed due to low replication are marked with 'NA', averages and errors are displayed where replication was available.

| Spring Attribute | | Number of Species | | | | | |
| --- | --- | --- | --- | --- | --- | --- | --- |
| | | 0 | 1 | 2 | 3–4 | >4 | |
| WETTED AREA (m$^2$) | Avg (±SE) | 16 (±4) | 382 (±221) | 446 (±172) | 1340 (±494) | 2426 (±929) | $p \leq 0.05$ |
| TOTAL POOL AREA (m$^2$) | Avg (±SE) | 1 (±0) | 151 (±88) | 122 (±61) | 573 (±319) | 184 (±122) | ns |
| MAXIMUM POOL DEPTH (mm) | Avg (±SE) | 1.4 (±0.8) | 7.1 (±2.1) | 8.8 (±2.3) | 13.4 (±2.7) | 13.8 (±3.0) | $p \leq 0.05$ |
| PH | Avg | 8.1 | 8.6 | 8.2 | 8.2 | 8.7 | ns |
| | (±SE) | NA | NA | (±0.1) | (±0.1) | (±0.2) | |
| COND. (uS) | Avg (±SE) | 875 NA | 802 NA | 1022 (±173) | 857 (±29) | 775 (±59) | ns |
| AVERAGE DISTANCE TO ALL SPRINGS (m) | Avg (±SE) | 63 (±10) | 111 (±32) | 124 (±45) | 150 (±49) | 77 (±12) | ns |
| AVERAGE DISTANCE TO CLOSEST 5 SPRINGS (m) | Avg (±SE) | 190 (±53) | 212 (±60) | 203 (±46) | 227 (±42) | 152 (±22) | ns |
| # OF SPRINGS WITHIN 300M | Avg (±SE) | 14.3 (±2.6) | 9.9 (±2.2) | 7.5 (±2.0) | 7.7 (±2.2) | 12.1 (±2.8) | ns |
| # SPRINGS CONNECTED BY WETLAND | Avg (±SE) | 0.1 (±0.1) | 0.2 (±0.1) | 0.3 (±0.1) | 1 (±0.4) | 0.8 (±0.3) | $p \leq 0.05$ |

## 5. Discussion

The six snail species had different environmental requirements, and different patterns of patch occupancy, and these differences were a combination of environmental restriction and apparent dispersal limitation. Springs with high diversity were those with a low distance to near neighbours (generally <300 m) and were connected via wetland to a significantly higher number of other springs. These results mirror those from previous studies in GAB springs in the south [19]. However, it is not valid to assume that environmental conditions are homogenous or ineffectual in dictating patterns of distribution as well. Gastropod species assessed here differ significantly from one another in the environmental conditions of the springs they occupy, as assimilated into Table 3. Some are restricted to the largest and deepest springs (e.g., *Glyptophysa* sp., *J. acuminata*); others occupy these as well as small springs with little more than a few meters of wetted sediment (e.g., *J. edgbastonensis*). The species that occupy the fewest springs are the most environmentally particular (i.e., are restricted to deeper larger springs), but also persist through time where multiple springs meeting their requirements are nearby—for example *Glyptophysa* sp. and *Gy. edgbastonensis* were restricted to springs with big pools but were found in both surveys in those springs that met this environmental requirement and had the nearest five neighbours within ~300 m.

**Table 3.** Summary of the number of springs on the Edgbaston Reserve section (~85 springs) of the Pelican Creek springs complex (~150 springs) occupied by each of the six focal species of gastropod endemic to the region, and the minimum measurements of spring size (represented as the total area of wetted sediment), pool size and maximum depth characteristics of the springs occupied by each species, and an estimate of the total wetland area of springs occupied by each species (where total wetland area is calculated from the wetted area estimate).

| Species | Number of Springs Occupied | Spring Size (m$^2$) | Pool Size (m$^2$) | Pool Depth (cm) | Total Wetland Area of OCCUPIED Springs (ha) |
|---|---|---|---|---|---|
| *Glyptophysa* sp. | 5 | >1000 | >10 | >10 | 2.8 |
| *Gy. edgbastonensis* | 10 | >100 | >5 | >5 | 3.2 |
| *J. acuminata* | 11 | >1000 | >5 | >5 | 4.0 |
| *J. jesswiseae* | 25 | >10 | >0 | >1 | 6.9 |
| *J. edgbastonensis* | 31 | >1 | ≥0 | 0 | 5.2 |
| *Ga. fontana* | 12 | >100 | >5 | >5 | 2.9 |

Despite dispersal limitation obviously playing a role to play in island-like systems such as desert springs, the role of environmental heterogeneity and the differential environmental requirements across the species concerned cannot be ignored. In spring systems from a range of contexts, environmental conditions such as depth [20,54], water chemistry and permanence [47,48,57], altitude [70], temperature [71,72] and disturbance gradient [73–77] are related to patterns of diversity and distribution. Springs provide a unique freshwater system with strong environmental gradients; areas of pool at the spring vent provide stable conditions with characteristics of the groundwater, whilst pool depth and the outflowing spring tail are strongly influenced by terrestrial conditions [1]. In this way, they are a unique from patch networks such as fragmented forests because the size of a patch covaries with other physio-chemical environmental drivers.

In the springs of Pelican Creek, pools >10 cm deep are the only areas that maintain environmental constancy in the face of the prevailing environmental variance dictated by their arid context [20]. Most species endemic to the GAB springs system are unable to withstand such variance—for example all species of gastropod considered here, bar one, perish within 6h of being dry [27] and similar patterns are found elsewhere [26], plants are most diverse in springs that have maintained constant conditions for long periods [56], and fishes rely on permanent pools around spring vents to persist through periods when springs retract [73]. Therefore, it is unsurprising that springs that provide such pools foster the highest diversity. The island-like nature of desert springs has led many to consider them through the lens of classical metapopulation or metacommunity theory. Such approaches assume environmental conditions are homogenous across patches [78], or that the environmental requirements of the species concerned are inconsequential, for they assume that differences in dispersal capabilities are the only processes dictating patterns of occupancy, diversity and persistence. The results presented here suggest that in GAB springs at least, models will benefit from considering environmental variance and the specific environmental requirements of the species in the system when attempting to predict species distributions or patterns of diversity.

This is not to say dispersal limits are unimportant, as narrow environmental requirements appear to be interacting with dispersal limitations to maintain the highly restricted distributions of most species. The nature of this interaction is species-specific, and appears also to vary across families. For example, all species within the family Tateidae (genus *Jardinella*) considered here have colonised, or have been found in, some of the most isolated springs (i.e., those with an average distance to the closest 5 springs >500 m), despite the fact that one of them (*J. acuminata*) has environmental limits equally strict as those of *Glyptophysa* sp. or *Gy. edgbastonensis*. For small gastropods such as these (all species of *Jardinella* are <5 mm total shell length), becoming entangled on animals and transported between more distant locations may be easier than for larger species (e.g., *Glyptophysa* sp. is on average twice that total length). Tateidae in other locations also appear able to disperse long

distances at regular intervals [19,22]. These avenues of dispersal in springs system are diverse and scale-dependent—active movement between near neighbours is facilitated by connected wetlands whilst opportune and random dispersal between distant springs can be facilitated by either phoresy or rafting on floodwaters. The results presented here reaffirm that the propensity for springs species to recolonise a spring from a near neighbour is an important driver of local diversity, and that different species endemic to springs utilise paths of dispersal in different ways [28]. As most previous studies have concerned species of Tateid, this is particularly important in spring complexes where more than one family are present.

Species endemic to springs are renowned for having highly restricted distributions [7,79,80]. Most species occupy GAB springs within a single complex, generally less than 50 km$^2$ [13]. The results presented here emphasise that, within their already limited distribution, the total amount of habitable area for many is precariously smaller still. For example, all species endemic to the Pelican Creek spring complex considered here have a global distribution of ~8000 ha, but the amount of spring fed wetland habitat within that represents only ~9 ha. As no species occupied every spring, the actual amount of wetland area occupied by any one of these species is as little as one-third of that (2.8–6.9 ha) (Table 3). Most species occupied <20% of springs, and these findings suggest that this is because only a subset of springs on site suit their requirements. This means no amount of assisted migration [81,82] can increase the area of habitat suitable for these species. Available habitat is restricted further still for many, as most occur in high abundance only in the deep pool area of the springs they occupy–an area that is often <50% of the total spring area [20]. Such severe restrictions make these species even more highly vulnerable to disturbances and threats than previously thought.

Despite the ongoing and ever-improving conservation efforts aimed at preserving GAB springs, threatening processes continue. Given the results presented here, even minor disturbances can have drastic effects. Habitat destruction by invasive ungulates [74–76,83–85], increased predation and changed trophic dynamics caused by introduced aquatic species [31,85], or the severe habitat loss caused by groundwater overdraft [3,5,84] have degraded large portions of springs habitat, and continue to do so. In this study, severe disturbances caused by pigs and cattle had occurred recently within 20% of springs, but this damage was concentrated in the larger more diverse springs. This is in the face of admirable and diligent efforts by those managing the property to remove pigs. For the most restricted species (*Glyptophysa* sp.), such disturbances had occurred in 43% of the occupied wetland area (Rossini unpub. data). There is also evidence that combinations of threatening processes are likely to compound with one-another to increase extinction probabilities. For example, the Red-finned Blue-eye (*S. vermeillipinnis*, a fish endemic to the Pelican Creek complex), like the environmentally-restricted species of snail assessed here, only persists in the pool areas of large deep springs [85]. The amount of pool area where this species can persevere has been physically reduced by drawdown of artesian pressure [84], which is further compromised by the invasion of the Western Mosquito fish (*Gambusia holbrooki*). This has led to a significant decline in occupied area [31], extirpation from all but one spring within its previously documented range [73,86], and severe population declines associated with cataclysmic loss of genetic diversity [87].

Unfortunately, such stories are not uncommon in springs in Australia [15,84,88] or elsewhere [1–6, 89]. Many species endemic to springs systems remain, to this day, at severe risk of extinction [7,80]. These results suggest that the precarious position of these unique species is not only a product of their limited abilities to move between springs, but for many, is also a result of their strict environmental requirements. The role of environmental variance in dictating distributions is species-specific [28]—the diverse range of taxa, life histories and evolutionary narratives within springs mean that there is no general rule regarding the relative importance of dispersal and environmental limitations applicable to all (even those that are closely related to one another), and the most diverse springs are those that match the unique requirements of each species simultaneously. To best preserve species in this unique system, we need actions that ensure large numbers of interconnected and environmentally

heterogenous springs persist through time. Actions targeted on helping species overcome dispersal limitation alone will not ensure their long-term persistence.

**Author Contributions:** Conceptualization, R.A.R. and G.H.W.; methodology, R.A.R.; formal analysis, R.A.R.; investigation, R.A.R.; resources, R.J.F.; data curation, R.A.R.; writing—original draft preparation, R.A.R.; writing—review and editing, R.J.F. and G.H.W.; visualization, R.A.R.; supervision, R.J.F. and G.H.W.; project administration, R.A.R.; funding acquisition, R.A.R. All authors have read and agreed to the published version of the manuscript.

**Funding:** This research was funded by a top-up scholarship awarded to R.A.R. by the Great Artesian Basin Coordinating Committee. We thank Bush Heritage Australia for hosting the researchers on-site at Edgbaston Nature Refuge.

**Acknowledgments:** We would like to acknowledge the Elders—past, present and emerging—of the Iningai and Bidjera peoples (Pelican Creek) and the Jagera and Turrbal peoples (Brisbane) as the custodians of the land we work upon. We thank the editors and two generous reviewers for their insight and patience. We thank Bush Heritage Australia, and their patrons, for their ongoing protection of Edgbaston, for allowing us access, and for the advice and assistance of their staff. This work would not have been possible without the volunteer assistance of UQ iROOS, Sasha Jooste, David 'Cujo' Coulton and Ian Rossini, and the statistical assistance of Luis Darcy Arregoitia Verde.

**Conflicts of Interest:** The authors declare no conflict of interest.

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
