# Peer review of "Different Species Requirements within a Heterogeneous Spring Complex Affects Patch Occupancy of Threatened Snails in Australian Desert Springs"

_water, doi:10.3390/w12102942_

Round 1
Reviewer 1 Report
This paper describes how environmental factors such as size, isolation and water measurements of desert springs in Australia influence occupancy and persistence of endemic spring snail species. Although well written, the methods, results, and figures are a bit difficult to follow and understand. These sections need the most work to improve the manuscript. Furthermore, the title doesn't match what the study is about. Probably should be closer to "Occupancy and persistence of threatened snails in Australian desert springs is affected by hydrological conditions and patch isolation" or something of that nature. Another suggestion is that species names need to be italicized throughout the manuscript. Also, it is not very clear why six species were chosen as a focal species when all of them could have been used or at least combine the families. More specific comments are below:
Line 9 "The" is in bold.
Line 11 what processes? environmental conditions? explain
Line 36 what environmental limits? explain
Line 65 km2 needs to be a superscript
Line 71 define zoochory
Line 82 citations should be at end of sentence
Line 94 define phoresy
Line 116 explain the range of spring systems. Also seems to be too many citations here
Line 120 what mediated processes?
Figure 1 is great. However what are the circled areas on the map on the left? Also add edgbaston reserve to the caption since reserve keeps being mentioned in the manuscript to make it easier to follow. Also add sample size to caption.
Line 59 references are not correctly formatted
Table 1 is not cited anywhere. This should also be in the results and remove ponder and rossini from title. In any case this should probably unnecessary since the information are in the figures 5 and 6.
Line 282 to 297. Why put them in diversity categories split into different bins with sample sizes of 12. Why not conduct a linear regression with the environmental conditions as predictors and species richness as response? This would give more easily interpretable results.
Table 2 is unreadable.
Line 330-339 perhaps better as a figure, or at least add averages and standard deviations to all the values.
Figure 2 is very low quality. I also don't think it shows much. Perhaps circle the ponds? I would just remove this figure as it is difficult to compare and interpret.
Figure 3 is good but the legend is hard to find. Perhaps better to place them within each subfigure. Also subscripts for km2, as well as for the following paragraphs.
Figure 4 is difficult to interpret especially the blue lines and the gray dots. Plus it seems to be the same as Figure 7 and 8. I would remove or add as supplementary.
Figure 5 and 6 should be combined into a-f
Figure 7 should be a supplementary
Figure 8 should come before before 5 and 6.
Table 3 needs to be in the results.
Line 520 similar patterns of reliance on permanent waterbodies and distance between them also found in amphibians https://doi.org/10.1016/j.biocon.2015.08.010
This paper describes how environmental factors such as size, isolation and water measurements of desert springs in Australia influence occupancy and persistence of endemic spring snail species. I think this would be a good fit for the journal and its readers. Although well written, the methods, results, and figures are a bit difficult to follow and understand. These sections need the most work to improve the manuscript. The title should also better match what the study is about. Furthermore, species names need to be italicized throughout the manuscript along with km2 being in superscript. Lastly, the figures and tables need to be improved. More specific comments are sent to the authors which should be taken into account.
Reviewer 2 Report
General comments:
The authors analyze how the geometry and local environmental conditions of freshwater spring “patches” affects their snail species composition and persistence. The authors surveyed numerous spring sites that had been previously sampled 7-9 years earlier, thus allowing species turnover to be assessed. Overall, the manuscript is quite well written, though the presentation of results is quite dense and hard to follow.
I have six major concerns with the current manuscript, which I feel should be addressed.
1) Although ecological metacommunity theory provides a very useful conceptual framework for this study, it is not mentioned. I recommend that the authors recognize this theory in their manuscript, because of its clear relevance to the general objective of this manuscript – i.e., to understand the relative roles of inter-patch dispersal and local habitat conditions of patches on community assemblages (as encapsulated in point 1 of their abstract), which are the ‘bread and butter’ of metacommunity theory (see e.g., Liebold & Chase 2017).
2) The methodology used for comparing the effects of patch geometry and environmental characteristics on snail species diversity does not allow one to determine which is more important, despite this being a central question of this study (see L 9-14). The authors should read López‐Delgado et al. (2019) who use a more appropriate methodology. As López‐Delgado et al. (2019) state “To examine the relative influence of dispersal and environmental variables on beta diversity and its components, distance-based redundancy analysis (db-RDA) and variation partitioning analysis were conducted”. As a result, López‐Delgado et al. (2019) were able to show that local environmental conditions can have greater influences than spatial factors on the species composition of local aquatic community assemblages.
3) In my opinion, the authors underestimate the amount of literature that already considers how local environmental conditions affect the species composition of springs. For examples of some older studies, see Glazier & Gooch (1987) and Gooch & Glazier (1991). For a recent review, also see Glazier (2014), which contains many relevant references conveniently classified in Table A1. In particular, this table lists 31 references dealing with how spring community assemblages relate to habitat type (going as far back as Bornhauser 1912, Beyer 1932 and Roll 1940). Other numerous references listed in Table A1 consider bedrock type, substrate type, disturbance, habitat persistence, temperature, water flow, water chemistry, etc. Many additional studies examining how the biotic composition of springs relates to local environmental conditions have appeared since 2014….
4) This manuscript could benefit from a fuller analysis of species turnover between the two sampling periods. In my opinion, this should be a highlight of this study, not a relatively minor, little analyzed and discussed aspect, as now treated in the present manuscript. See also specific comments.
5) It would be helpful if the authors clearly explained how the present study differs from two previously published studies that also examine how local environmental conditions of various Australian artesian springs affect the occurrence of snail species (i.e., Rossini et al. 2017a, b).
6) This manuscript is not properly formatted for the journal Water (MDPI). For example, the references should be numbered in the text and References section. The sections of the manuscript should also be numbered. Etc…
Specific comments:
L 9: Remove bold letters for “The”.
L 29, 314: Change “sub-set” to “subset”.
L 51-53: See also Glazier (2014).
L 71-73, 89-94, 249-251, 309-311, etc.: Run-on sentences.
L 90: Please italicize genus and species names here and elsewhere.
L 94: I suggest changing “Each species ability” to “The ability of each species”.
L 112-115: At some point in the Introduction it would be useful if the authors pointed out that effects of dispersal and local habitat conditions (habitat heterogeneity) are both important components (among other factors) of current metacommunity theory (see e.g., Leibold & Chase 2017). Surprisingly no mention of metacommunity theory or the metacommunity concept is made, despite its clear relevance to the present manuscript. Metacommunity theory provides a nice conceptual framework for this study.
L 115-119: See also Glazier & Gooch (1987) who show how macroinvertebrate assemblages match with local spring environments using methods of ordination analysis similar to that of the present study.
L 127: Change “affecting” to “affects”.
L 132-135: See also Gooch & Glazier (1991) and other references cited in Glazier (2014) which discuss how the size and persistence of springs may affect their community assemblages.
L 280: Possibly change the awkwardly worded “that have stayed the same, seen an extirpation or colonization…” to “that have persisted, disappeared or newly arrived…”?
Table 2: This table cannot be entirely read because it extends beyond the margins of the page.
L 314-318: Here, the authors document species turnover in their study springs without explicitly saying so. It would be interesting if they calculated mean percentage species turnover in their study system, and how turnover rate varied with the size, isolation and environmental characteristics of the springs sampled. I know of no other study of spring community assemblages that has done this. Why is this important aspect of this study so little analyzed and discussed?
Figure 2: The pictures are quite fuzzy. Are more focused pictures available?
L 504-510: See also Glazier & Gooch (1987), Gooch & Glazier (1991), and Glazier (2014).
L 563-568: See also Glazier (2014).
Discussion (L 481-584): Unfortunately the central question of this study – i.e., whether dispersal or local environmental conditions are more important in determining local species diversity is not answered clearly. Therefore, I found the discussion to be quite disappointing.
Literature cited:
Beyer, H. (1932). Die tierwelt der quellen und bache des baumbergegebietes. Abhandlungen aus dem Westfälischen Provinzial-Museum für Naturkunde 3, 9–187.
Bornhauser, K. (1912). Die tierwelt der quellen in der umgebung Basels. Internationalen Revue der Gesamten Hydrobiologie und Hydrographie, Biologische Supplemente 5, 1–90.
Glazier, D.S. (2014) Springs. Pages 1-78 In: Reference Module in Earth Systems and Environmental Sciences, ed. Elias, S.A. Waltham, MA: Elsevier.
Glazier, D.S., & Gooch, J.L. (1987). Macroinvertebrate assemblages in Pennsylvania (USA) springs. Hydrobiologia, 150(1), 33-43.
Gooch, J.L., & Glazier, D.S. (1991). Temporal and spatial patterns in mid-Appalachian springs. Memoirs of the Entomological Society of Canada, 123(S155), 29-49.
Leibold, M.A., & Chase, J.M. (2017). Metacommunity Ecology (Vol. 59). Princeton University Press.
López‐Delgado, E.O., Winemiller, K.O., & Villa‐Navarro, F.A. (2019). Local environmental factors influence beta‐diversity patterns of tropical fish assemblages more than spatial factors. Ecology, e02940.
Roll, H. (1940). Weitere waldquellen Holsteins und ihre pflanzengesellschaften. Soziologisch-limnologische quellenstudien 2. Archiv für Hydrobiologie 36, 424–465.
The authors pose an important question, but do not answer it adequately because of inadequate methodology. This and other problems cause me to recommend major revision before this study is considered for publication. I hope that my comments help during the revision process. I think that this manuscript has the potential to be much better.
Round 2
Reviewer 1 Report
Great work on the manuscript it has drastically improved.
Author Response
Thank you
Reviewer 2 Report
I appreciate the valuable results that the authors have presented, but the present revised manuscript still remains misleading in two important ways.
1) Although the authors now mention metacommunity theory in their manuscript, they misrepresent it by saying in their response that it assumes “that differences in dispersal capabilities are the only processes dictating patterns of occupancy, diversity and persistence.” The authors make similar statements in their revised manuscript. This is patently false. Metacommunity theory explicitly considers not only dispersal (regional controls), but also biotic interactions and various environmental features of local habitats (local controls) in determining the presence or absence of specific species in local communities. I stick by my opinion that the authors should read and cite pertinent studies in metacommunity theory (e.g., Liebold & Chase 2017; Thompson et al. 2020), as I previously recommended. Then they will be able to place their study in an appropriate theoretical context. As a recent application of metacommunity theory to aquatic communities, focusing on both spatial and environmental factors, see Mozzaquattro et al. (2020).
2) The authors’ manuscript gives the misleading impression that their study’s focus on how local environmental features of local springs affect the presence/absence of specific species and thus their faunal composition represents a little used approach. The authors state in their abstract (point 2): “Many have strict environmental requirements, but the role of environmental heterogeneity amongst springs has rarely been considered alongside conventional patch characteristics (isolation and patch geometry)”. I would argue the opposite, at least for spring community studies! The reason why I cited so many references showing an emphasis on local environmental controls in my original review is to show the authors that they are mistaken. I can appreciate space limits on number of references, but selective citing of references (including arbitrary exclusion of older references simply because they are older) can be misleading. The authors do not even bother to cite a recent review of springs that cites numerous (> 30) relevant studies about environmental effects on local spring communities (Glazier 2014).
My comments are intended to help the authors place their useful results in a more general theoretical and historically appropriate context.
Literature cited:
Glazier, D.S. (2014) Springs. Pages 1-78 In: Reference Module in Earth Systems and Environmental Sciences, ed. Elias, S.A. Waltham, MA: Elsevier.
Leibold, M.A. & Chase, J.M. (2017). Metacommunity Ecology. Princeton University Press, Princeton, NJ.
Mozzaquattro, L.B., Renato Bolson Dala-Corte, R.B., Becker, F.G. & Melo, A.S. (2020). Effects of spatial distance, physical barriers, and habitat on a stream fish metacommunity. Hydrobiologia 847, 3039–3054.
Thompson, P.L., Guzman, L.M., De Meester, L., Horváth, Z., Ptacnik, R., Vanschoenwinkel, B., Viana, D.S. & Chase, J., (2020). A process-based metacommunity framework linking local and regional scale community ecology. Ecology Letters https://doi.org/10.1111/ele.13568.
Author Response
We thank the reviewer for their comments and their dedication to the MS. We understand that the attitude of this reviewer is that this MS is lacking in references and lacking in depth.
We believe the depth of resources cited is sufficient; at the editors discretion if this will block publishing we will insert. The attitudes of the authors regarding approaches to spring research missing research or misleading we do not agree. We have included a broad range of springs research from a broad range of contexts. They have recommended a book, we have included books and older resources. Again, at the editors discretion we will insert if requested.